# Highly CO Selective Trimetallic Metal-Organic Framework Electrocatalyst for the Electrochemical Reduction of CO$_2$

Tran-Van Phuc [ID], Jin-Suk Chung and Seung-Hyun Hur *

School of Chemical Engineering, University of Ulsan, Daehak-ro 93, Nam-gu, Ulsan 44610, Korea; vanphuc0509@gmail.com (T.-V.P.); jschung@ulsan.ac.kr (J.-S.C.)
* Correspondence: shhur@ulsan.ac.kr

**Abstract:** Pd, Cu, and Zn trimetallic metal-organic framework electrocatalysts (PCZs) based on benzene-1,3,5-tricarboxylic were synthesized using a simple solvothermal synthesis. The as-synthesized PCZ catalysts exhibited as much as 95% faradaic efficiency towards CO, with a high current density, low onset potential, and excellent long-term stability during the electrocatalytic reduction of CO$_2$.

**Keywords:** metal-organic framework; electrocatalyst; electrochemical reduction of CO$_2$; CO selectivity; stability

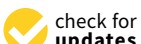

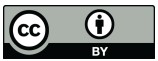

## 1. Introduction

The expanding quantity of CO$_2$ emissions has also created significant environmental issues such as global warming and severe weather, and hence considerable attempts have been made to decrease the amount of CO$_2$ in the atmosphere in order to address these problems. Of the different approaches, the electrochemical reduction of CO$_2$ (ERC) is one of the best since CO$_2$ can be transformed by the ERC process into useful chemicals such as alcohols, hydrocarbons, and formic acids [1–3].

The designing of efficient electrocatalysts is a key factor in the ERC. Recently, metal-organic frameworks (MOFs) have been widely studied in the areas of separation [4–8], catalysis [9–13], sensors [14,15], water treatment [16–18], and drug delivery [19–21] because their diversity of metal ion and organic ligand combinations result in porous structures. MOFs also can be a good candidate for the electrode materials of the ERC due to their large surface area and excellent CO$_2$ adsorption capability [22–25].

Noble metals have been widely used as effective electrocatalysts because of their high catalytic activity and excellent stability. However, their high cost and non-abundance in the earth are regarded as huge disadvantages in terms of their usage for the economical ERC electrode materials [26–28]. Recent studies have shown that non noble metals such as Cu and Zn components could be good candidates for the ERC electrode materials because of their high electrocatalytic activity and high abundance in the earth, but it still needs improvement in terms of CO selectivity [29–35].

In this study, benzene-1,3,5-tricarboxylic (BTC)-based Pd, Cu, and Zn trimetallic MOF electrocatalysts (PCZs) were synthesized with a simple solvothermal synthesis. Increases in ERC activity, low onset potential, low Tafel slope, and an outstanding selectivity of CO inside ERC were greatly improved with the addition of Pd to the Cu and Zn bimetallic MOF. The findings showed that up to 95% of faradaic CO efficiency can be obtained by using an optimized PCZ as an ERC electrode, which was around two times the bimetallic-BTC value (≈49%) and the comparable value of previously recorded ERC electrodes, as outlined in Table S1. The increased area of surface and electron transfer from Pd to Cu by adding Pd could lead to improved ERC properties.

## 2. Result and Discussion

### 2.1. Characterization of the PCZs

The morphology and microstructure were characterized by magnified field emission scanning electron microscope (FE-SEM) and transmission electron microscope (TEM). As shown in Figure 1, all PCZs exhibited porous morphology but PCZ-2 showed the most porous structures among as-synthesized PCZs. The $N_2$ adsorption–desorption isothermal experiment was conducted to analyze the specific surface area and pore structure of as-synthesized PCZs and BM-3 (Figure S1). The specified surface area and the pore volume of PCZ-1, PCZ-2, PCZ-3, and BM-3 were determined to be 126.11, 271.40, 40.74, and 22.15 $m^2g^{-1}$, and 0.32, 0.45, 0.11, and 0.11 $cm^3g^{-1}$, respectively (Table S2). The PCZ-2 exhibited the largest specific surface area and pore volume among all catalysts, which was in good agreement with SEM results.

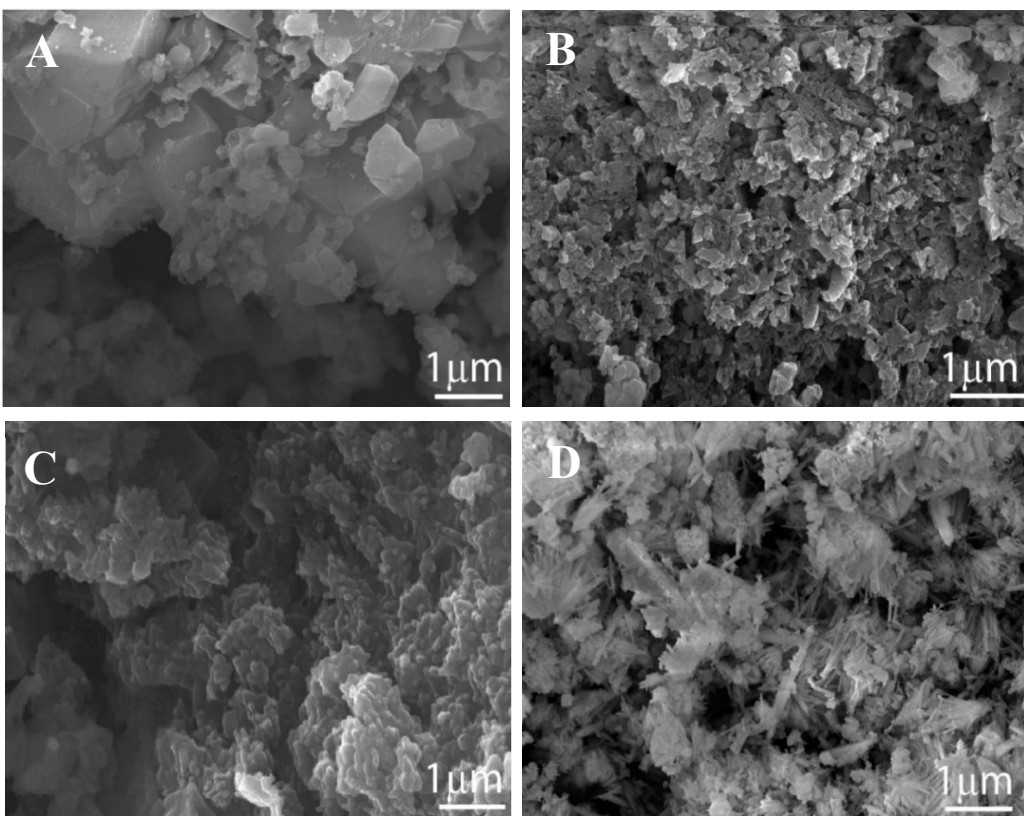

**Figure 1.** The SEM images of (**A**) PCZ-1, (**B**) PCZ-2, (**C**) PCZ-3, and (**D**) BM-3.

The TEM images indicated the uniformly distributed metal components of as-synthesized PCZs (Figure 2). In addition, we could clearly observe the lattice fringes of Cu (111), Zn (002), and Pd (111) in the high-resolution transmission electron microscope (HR-TEM) images of as-synthesized the PCZs. It is worth noting that three metal components were very close to each other instead of isolating, which indicated possible interactions among them.

The X-ray diffraction (XRD) was performed to investigate crystal structures of the as-synthesized catalysts. As shown in Figure 3, all PCZs exhibited the clear MOF patterns of Cu-BTC (JCPDS 00-064-0936), which indicated the successful formation of MOF structures of as-synthesized PCZs.

In addition, Pd, Cu, and Zn metallic peaks were dominated for all PCZs instead of metal oxide-related peaks in the BM-3 (Figure S2), which revealed the interaction among metal components during the synthesis of PCZs.

The X-ray photoelectron spectroscopy (XPS) was used to analyze the chemical composition and oxidation status of as-synthesized PCZs and BM-3. As shown in the XPS survey

spectra (Figure 4A), metal components of Pd, Cu, and Zn were clearly observed in all the PCZs, but only Cu and Zn were observed in the BM-3 (Figure S3).

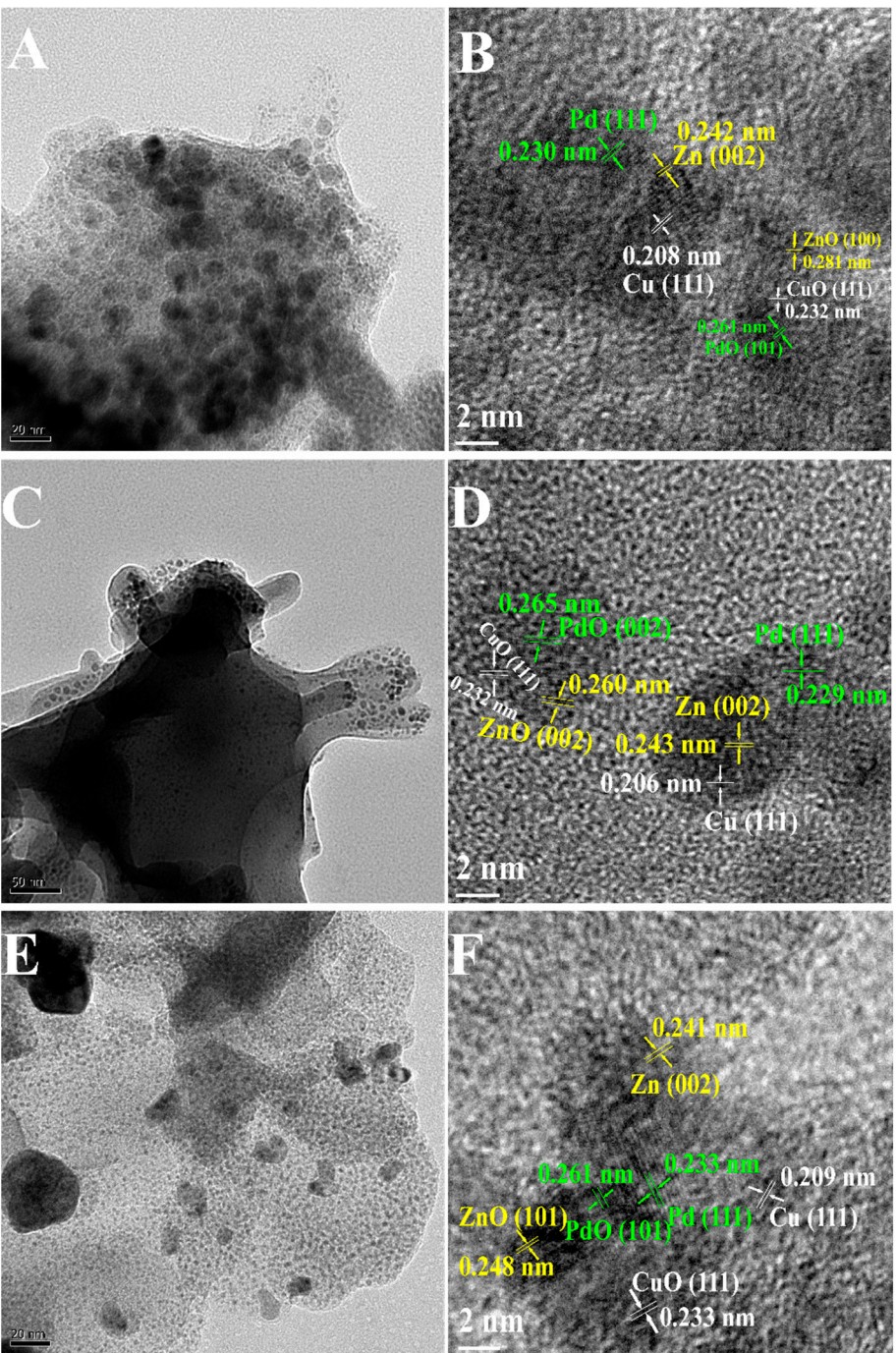

**Figure 2.** The TEM and HR-TEM images of (**A**,**B**) PCZ-1, (**C**,**D**) PCZ-2, and (**E**,**F**) PCZ-3.

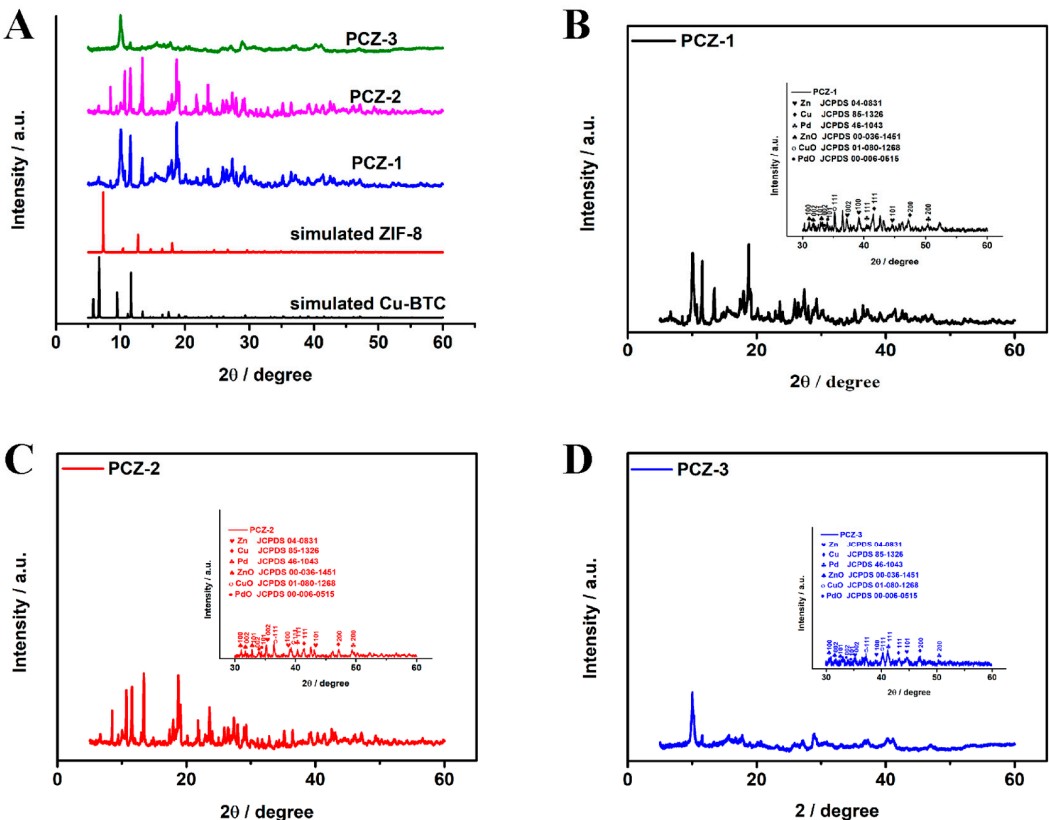

**Figure 3.** (**A**) XRD patterns of PCZs and simulated XRD patterns of ZIF-8 and Cu-BTC. XRD and PXRD (inset) patterns of PCZs (**B–D**).

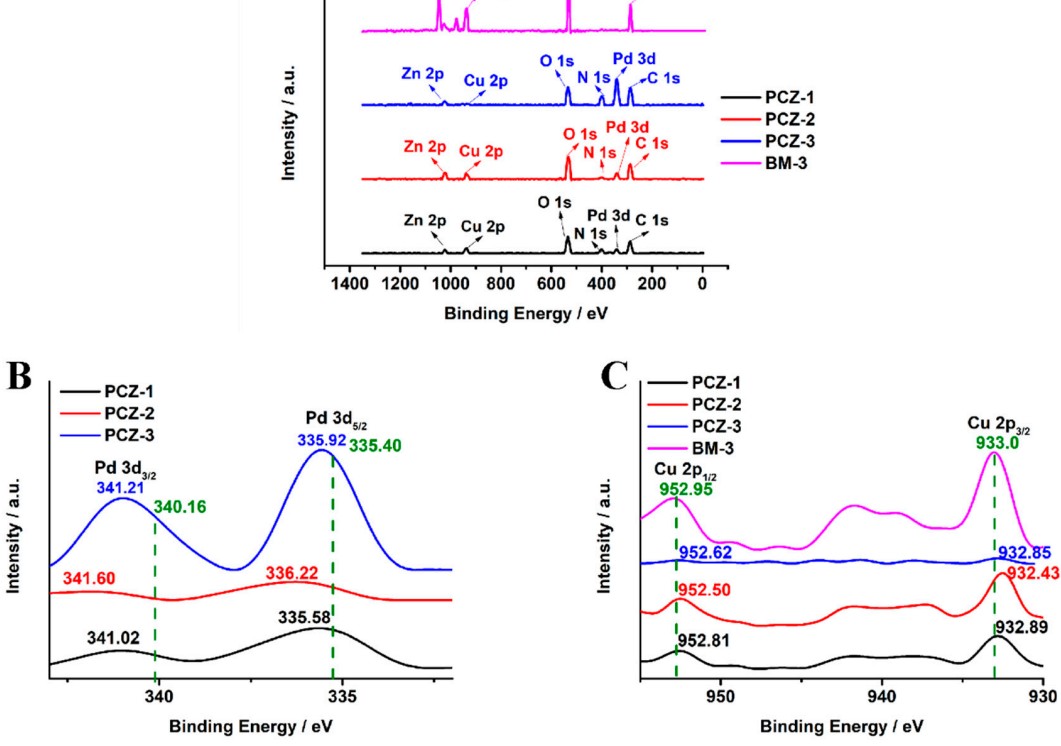

**Figure 4.** (**A**) XPS survey spectra, (**B**) Pd 3d, and (**C**) Cu 2p of PCZs and BM-3 (the dashed line ( ⋯ ) marks the peak position of monometallic Pd and Cu).

As shown in Figures S4 and S5, the Zn 2p peaks of PCZs were observed at 1021 and 1046 eV, which can be assigned to Zn 2p$_{3/2}$ and Zn 2p$_{1/2}$, respectively, indicating that Zn is in the ionic state in PCZs [36,37]. Two range peaks at the binding energies of (933.7 to 935.1 eV) and (952.8 to 954.7 eV) can be attributed to the Cu 2p$_{3/2}$ and Cu 2p$_{1/2}$, respectively [13,38]. The two pairs of Pd 3d signals of PCZ-2 were observed at (341.6 to 343.4 eV, Pd 3d$_{3/2}$) and (336.5 to 337.8 eV, Pd 3d$_{5/2}$), which can be assigned to the Pd and PdO species, respectively [39]. It has been observed that Pd has selectivity towards CO formation; hence, with the increased Pd/Cu ration, the product selectivity towards CO would increase. It is interesting to note that the positions of both the Pd 3d and Cu 2p peaks shifted with the change in ratio in the PCZs (Figure 4B,C). These opposite shifts can be attributed to the electron transfer between Pd and Cu atoms during catalyst formation. The different surface composition might result in different CO selectivity [40–42]. It was reported that different Cu to Pd ratio leads to the different electrocatalytic activity due to the different orientations of the intermediates on the surface and changing the chemisorption energy of reactant molecules on the surface of Pd atoms by the effect of the different surface composition and elemental distribution of Pd and Cu on the surface [43]. We believe that this kind of synergetic effect between Pd and Cu might enhance the ERC activity of as-synthesized PCZ catalysts.

*2.2. ERC Properties of PCZs and BM-3 Catalysts*

Figure 5A shows the electrochemical response of PCZs and BM-3 performed by linear sweep voltammetry (LSV) with the potential range from −0.35 to −1.85 V vs. RHE. With the addition of Pd, the current density increased significantly, and PCZ-2 exhibited best ERC activity among them. In addition, the positive shift of onset potential was also observed, and PCZ-2 showed about 235 mV more positive onset potential than that of BM-3. We believe that large BET surface area and electronic interaction between Pd and Cu of PCZ-2 may result in enhanced ERC properties. With the increased ratio of Pd ions, catalytic performance became better due to the increased porosity in the structure, and with the small amount of Pd in catalyst, electrocatalytic ability decreased as the holes collapsed in the structure [44,45]. The chronoamperometry test conducted at a constant potential (−0.75 V vs. RHE) and Tafel slope also showed similar results with that of LSV experiments. The PCZ-2 exhibited the highest current density and lowest Tafel slope among all catalysts with stable output (Figure 5B,C).

The electrochemical active surface area (ECSA) was measured by cyclic voltammetry (CV) at a potential ranging from 0.75 to 0.85 V vs. RHE. The ECSA was obtained by the equation of ECSA = $C_{dl}/C_s$, where $C_{dl}$ is the slope of the linear regression between the current density difference and the scan rates, and $C_s$ is the specific capacitance of the standard electrode material on a unit surface area [1,46,47]. As shown in Figure S5, the ESCA value of PZC-2 (311.72 $cm^2_{ECSA}$) was about one order higher than those of PCZ-1 and PCZ-3 (9.10 $cm^2_{ECSA}$ and 0.123 $cm^2_{ECSA}$, respectively), which might lead to an excellent ERC activity of PCZ-2.

To confirm that the electrocatalytic current was originated from the $CO_2$ reduction, we performed the electrocatalytic reaction under $CO_2$ and $N_2$ atmospheres. As shown in Figure 5D, only negligible current was observed at around −0.85 V vs. RHE under the $N_2$, which indicates that the largest part of the electrochemical current of PCZ-2 near the onset potential came from the $CO_2$ reduction. The current increase at high potential was due to the hydrogen evolution reaction.

The faradaic efficiency (FE) for different products during the ERC under the various potential was evaluated by analyzing the gas and liquid phase components with gas chromatography and $^1$H NMR, respectively. As shown in Figure 6, PCZ-2 catalyst exhibited as high as 95% FE for CO at −0.75 V vs. RHE, which was higher than any other FEs obtained by other catalyst value. For all tested catalysts, the sum of FE of $H_2$ and CO was approximately 100% under the potential range from −0.25 V to −1.05 V vs. RHE. No other product in the liquid phase could be detected by $^1$H NMR, even after 10 h chronoamperometric test

(Figure S6), which revealed that only CO and $H_2$ could be obtained with high CO selectivity when PCZ-2 was used as an ERC electrode. It was observed that Pd has selectivity towards CO formation; hence, with the increased Pd/Cu ration, the product selectivity towards CO would increase. The different surface composition might result in the different CO selectivity [48,49].

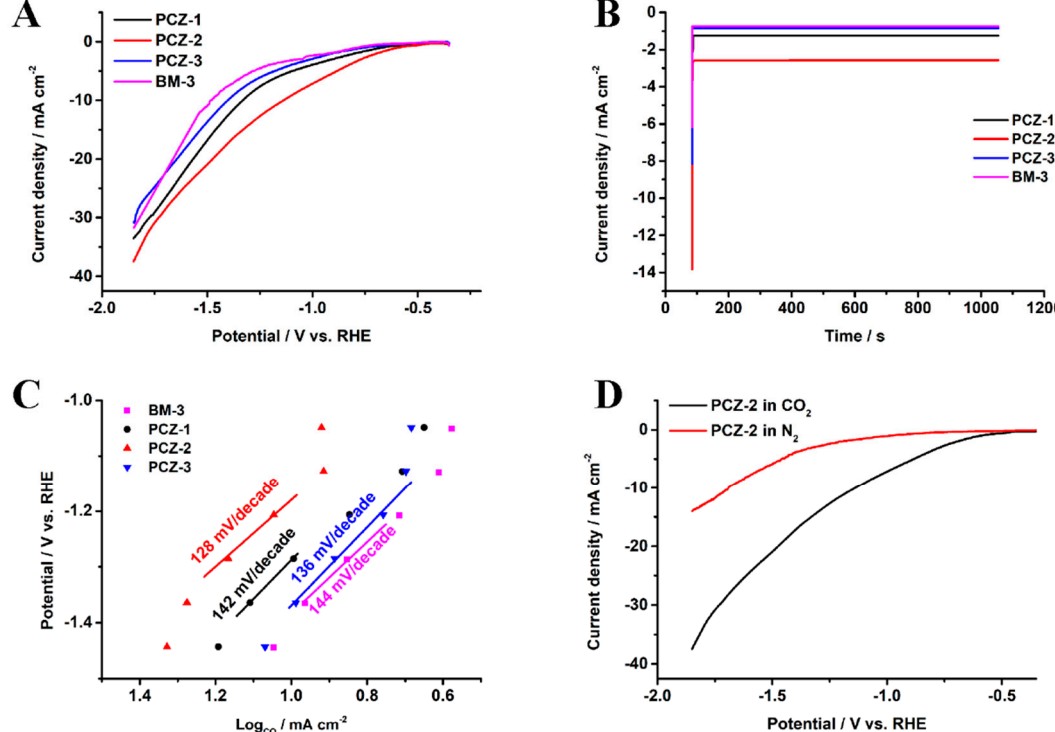

**Figure 5.** (**A**) LSV curves and (**B**) chronoamperometry PCZs and BM-3 at −0.75 V vs. RHE. (**C**) Tafel plots with a linear fit of PCZs and BM-3 catalysts. (**D**) LSV curves of PCZ-2 in $N_2$ and $CO_2$. We used 0.1 M $KHCO_3$ aqueous solution as an electrolyte for all experiments.

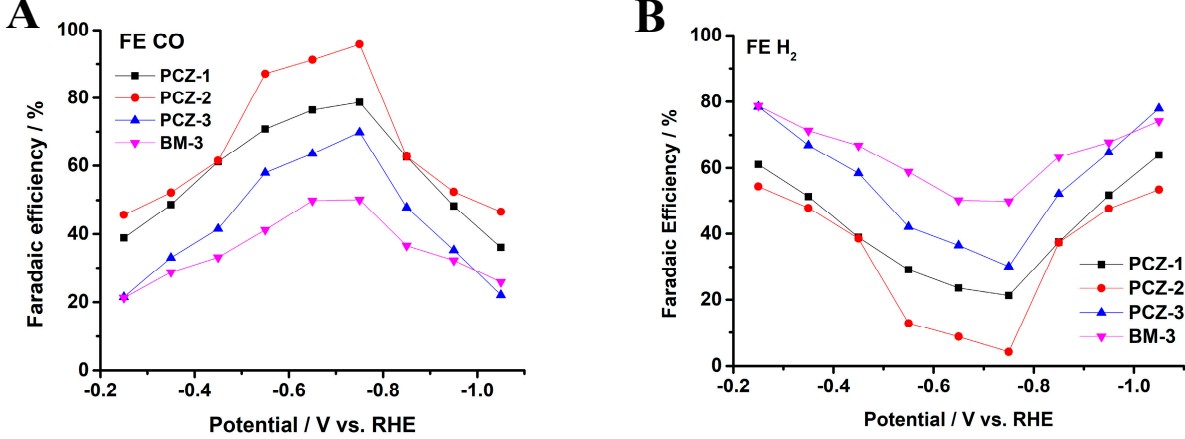

**Figure 6.** The faradaic efficiency (FE) for (**A**) CO and (**B**) $H_2$ of the PCZs and BM-3 catalysts at various potentials.

The stability of the PCZ-2 catalyst was evaluated by a 10-h chronoamperometry test conducted at a constant potential (−0.75 V vs. RHE). As shown Figure 7, both the current density and FE of CO were almost constant even after 10 h ERC operation, which indicates the excellent stability of as-prepared PCZ-2 towards ERC.

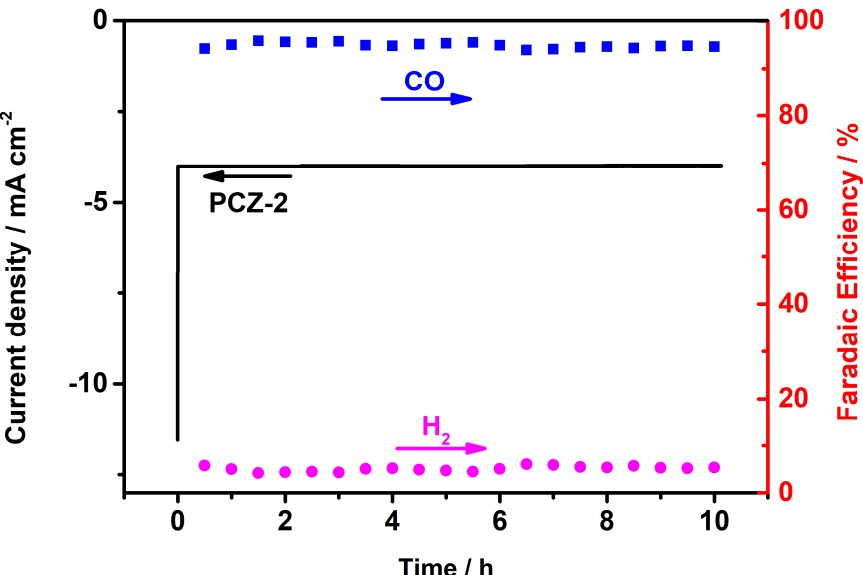

**Figure 7.** Chronoamperometric long term stability test of PCZ-2 during the electrochemical reduction of $CO_2$ process for long-term test.

To explore the stability of PCZ-2 further, we analyzed the PCZ-2 catalyst with XRD, SEM, and XPS after a 10-h chronoamperometry test. As shown in Figures S7–S10, no obvious changes in the morphology, electronic state, and crystal structure were observed, which also indicates an excellent stability of as-prepared PCZ-2 catalyst.

## 3. Materials and Methods

### 3.1. Materials

Zinc nitrate hexahydrate ($Zn(NO_3)_2·6H_2O$, 99%), 2-methylimidazole (2-MeIm, 99%), palladium(II) chloride ($PdCl_2$,99.9%), copper (II) nitrate trihydrate ($Cu(NO_3)_2·3H_2O$, 99%), benzene-1,3,5 tricarboxylic acid (BTC, 95%), N,N′-dimethylformamide (DMF) (99.5%), methanol ($CH_3OH$, ≥99.9%), and ethanol ($C_2H_5OH$) were purchased from Sigma-Aldrich. Throughout this job, DI water was used.

### 3.2. Characterization Techniques

The XRD analysis was carried out on a (D/MAZX 2500 V/PC model, Rigaku, Japan) using Cu K radiation (40 kV, 30 mA, λ = 1.5415). Thermo ESCALAB 250 Xi, Thermo Fished Scientific of the USA, with Al Kα X-ray radiation (1486.6 eV) was used to collect the catalyst X-ray photoelectron spectroscopy (XPS). PCZ powder morphologies were performed by means of field-emission electron microscopy (FE-SEM, JEOL JSM-600F, Tokyo, Japan). A thermal conductivity detection (TCD) and a column of HP-PLOT U were injected into a gas chromatograph (YOUNG LIN-Acme 6100 GC, Anyang, Korea). Liquid goods at room temperature were distinguished by $^1$H NMR spectrometer (400.13 MHz for 1H, 100.61 MHz for 13C). An electrochemical workstation was used to electrochemically reduction $CO_2$ (Bio-Logic, SP-50, Knoxville, TN, USA).

### 3.3. Synthesis of ZIF-8 and Pd, Cu, and Zn Trimetallic MOF Electrocatalysts (PCZs)

#### 3.3.1. Synthesis ZIF-8

Typically, $Zn(NO_3)_2·6H_2O$ (3 g) and 2-MeIm (7 g) were dissolved in 200 mL methanol separately under magnetic stirring for 15 min for each solution. Both formulas blended quickly for 24 h under magnetic stirring. At 10,000 rpm, the mixture was separated in centrifugation for 5 min. Finally, the sample was collected and washed 3 times under a vacuum with methanol (30 mL) and dried for 12 h at 80 °C.

### 3.3.2. Synthesis of PCZs

The PCZs were synthesized via a solvothermal process by the following the previous report [50] with some modifications, as shown in Scheme 1. In brief, ZIF-8 was synthesized following the procedure reported by Chen et al. with some modifications [51]. Typically, $Cu(NO_3)_2$, $PdCl_2$, and BTC were dissolved into the 45 mL mixed solution of $H_2O$/ethanol/N,N′-dimethylformamide (DMF) and stirred in the air for 15 min with magnetic bar, and the PCZs were obtained after reaction time. Then, the solution was transferred to Teflon-autoclave and was kept at 85 °C for 20 h. The final product was filtered and washed 3 times with methanol. Finally, it was dried at 90 °C 24 h in the vacuum oven. The samples were named as PCZ-n (n = 1–3) depending on the molar ratio of Pd/Cu/Zn. The molar ratios of PCZ-1, PCZ-2, and PCZ-3 were 0.25:0.75:1, 0.5:0.5:1, and 0.75:0.25:1, respectively. For the comparison, Cu and Zn bimetallic electrocatalysts (BM-3) were also synthesized by the same procedure with the 1:1 molar ratio of Zn to Cu.

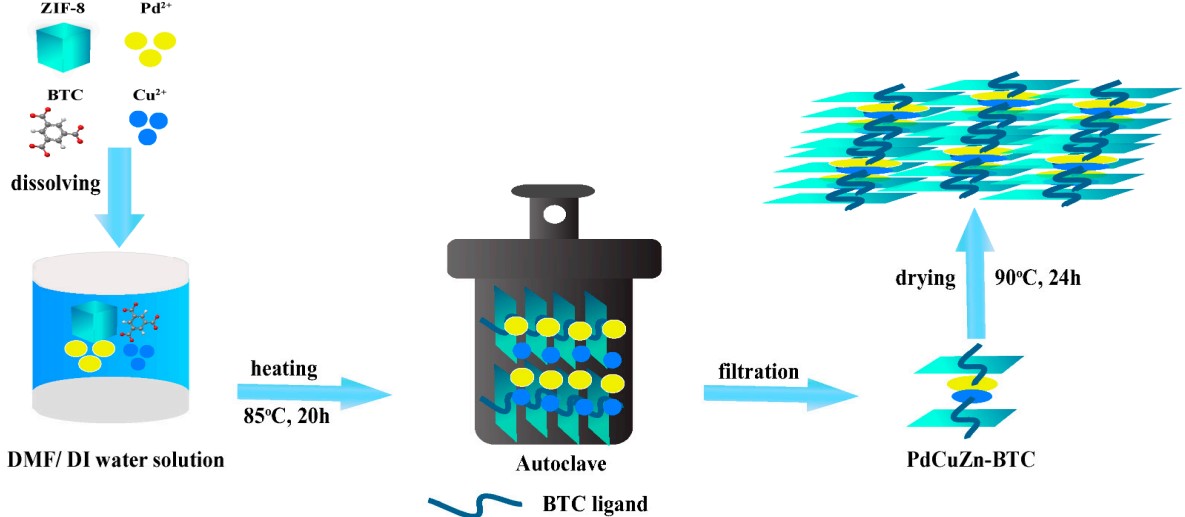

**Scheme 1.** The schematic illustration of PCZ synthesis.

### 3.4. Preparation of the Working Electrodes

First, 5 mg of PCZs or BM-3 was weighted and suspended in a 3 mL Nafion/isopropyl alcohol mixture solution. The ultrasonic bath (Jeio-Tech Co., Daejeon, Korea) was utilized to blend the dispersant thoroughly for a total of 15 min to ensure dispersion of the mixture before application to the electrode (2 mm diameter). The drying chamber was made hotter with a hot air gun for 10 minutes before the electrode (volume = 0.079 $mg/cm^2$) was used up.

### 3.5. Electrochemical Measurements

The catalysis of the PCZs and BM-3 was carried out using a linear sweep voltameter (LSV) with a potential range of −0.35 to −1.85 V vs. RHE with 0.1 M $KHCO_3$ that was $CO_2$-saturated at a scan rate of 20 mV s$^{-1}$. Using an H-type cell, we expanded the cells, wherein the electrolytes had been placed on one side, with a Nafion 115 membrane separating the anode and cathode compartments. In order to ensure the electrodes had no harmful interactions, we had cathodic and anodic compartments, one for the glassy carbon electrode (GCE) and the standard calomel reference electrode (SCE), and one for a counter electrode made of Pt mesh. The electrochemical reduction reactions were carried out at room temperature in a 0.1 M $KHCO_3$ aqueous solution. Prior to ERC, $CO_2$ was continuously bubbled into the cathode compartment at a flow rate of 2 mL min$^{-1}$ for 30 min to saturate the 0.1 M aqueous $KHCO_3$ electrolyte. Potentials versus RHE (reversible hydrogen electrode) were transformed into SCE (saturated calomel electrode) according to the Nernst equation, with pH value = 6.9 ($E_{RHE}$ = $E_{SCE}$ + 0.244 + 0.059 * pH). The excess

capacity was used at room temperature and 1 atm for gaseous materials from $-0.25$ V to $-1.05$ V (vs. RHE). A gas chromatographer examined the outlet gas product each 25 min (YOUNG LIN-Acme 6100 GC, Anyang, Korea).

## 4. Conclusions

The Pd, Cu, and Zn trimetallic metal-organic framework electrocatalysts (PCZs) were synthesized using a simple solvothermal synthesis. The optimally synthesized PCZ catalyst exhibited a very high CO selectivity with excellent ERC properties, including low overpotential, high current density, and long-term stability. The increased surface area, electrochemical active surface area, and the electronic interaction by the addition of Pd enhanced the electrocatalytic activity and faradaic efficiency of the Cu and Zn metal-organic framework electrocatalyst towards CO.

**Supplementary Materials:** The following are available online at https://www.mdpi.com/article/10.3390/catal11050537/s1, Figure S1: $N_2$ adsorption–desorption isothermal curves of PCZs and BM-3. Figure S2: XRD patterns of BM-3. Figure S3: The XPS spectra (a) Cu 2p and (b) Zn 2p of BM-3. Figure S4: The XPS spectra Pd 3d, Cu 2p, and Zn 2p of (a,d,g) PCZ-1, (b,e,h) PCZ-2, and (c,f,i) PCZ-3. Figure S5: The measured capacitive currents as a function of scan rate for PCZ catalysts. Figure S6: $^1$H-NMR spectra of the electrolyte after $CO_2$ reduction electrolysis at -1.4 V vs. SCE for long-term test. Figure S7: XRD patterns of PCZ-2 after long-term test. Figure S8: The SEM image of PCZ-2 after long-term test. Figure S9: The XPS spectra (a) Pd 3d, (b) Cu 2p, and (c) Zn 2p of PCZ-2 after long-term test. Figure S10: XPS survey spectra of PCZ-2 after long-term test. Table S1: CO selectivity using among various catalysts. Table S2: Surface area and pore size of PCZ-n (n= 1–3) and BM-3 samples.

**Author Contributions:** Conceptualization, T.-V.P and S.-H.H.; methodology, T.-V.P.; validation, J.-S.C. and S.-H.H.; formal analysis, T.-V.P.; investigation, T.-V.P.; resources, J.-S.C. and S.-H.H.; writing— original draft preparation, T.-V.P.; writing—review and editing, J.-S.C. and S.-H.H.; supervision, J.-S.C. and S.-H.H.; project administration, J.-S.C. and S.-H.H.; funding acquisition, J.-S.C. and S.-H.H. All authors have read and agreed to the published version of the manuscript.

**Funding:** This study was supported the National Research Foundation of Korea (NRF) grant funded by the Korean government (MSIT) (2020R1A4A4079954).

**Data Availability Statement:** Data available on request.

**Conflicts of Interest:** The authors report no declarations of interest.

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
