# Peer review of "Highly CO Selective Trimetallic Metal-Organic Framework Electrocatalyst for the Electrochemical Reduction of CO2"

_catalysts, doi:10.3390/catal11050537_

Round 1

Reviewer 1 Report

The reviewed manuscript describes the investigation of trimetallic (Pd-Cu-Zn) metal-organic framework as electrocatalyst for electrochemical reduction of CO2. The study covers the synthesis of PCZs with different atomic ratios between Pd, Cu and Zn, their morphological studies via SEM and TEM, surface characterization using XPS and electrochemical experiments (LSV, CA) with synthesized specimens. In general, the manuscript is interesting from scientific point of view and logically written.

I recommend the manuscript to be published in Catalysts only in case of proper major revisions:

  1. The literature overview on Cu-based electrocatalysis for ERC of CO2 is very limited. There are many groups, which extensively study the Cu-based ERC electrocatalysts (e.g. group of Prof. Cuenya, ACS Catal. 9 (2019) 5496; Nature Commun. 11 (2020) 3489; J. Am. Chem. Soc. 136 (2014) 6978, etc.). The overview need to be completed.
  2. What is the reason for such composition choice? Why Pd:Cu:Zn = 1:1:1? Why specimen with component ratios Pd:Cu:Zn = 1:1:3 was not synthesized?
  3. The comparison of PCZ-2 activity with other known electrocatalysts is presented in Table S1 and potential is written in V vs. RHE, whereas throughout the manuscript it is in V vs. SCE. The units need to be unified and better to present all values and graphs in V vs. RHE.
  4. Why the sampling by GC was done every 25 min (so rarely)?
  5. More details and clarifications are necessary for Table S1: what is the meaning of column “Applied potential”? Onset potential? Potential to reach the mentioned current density? Or?? The same question is about so different values of current densities? How electrocatalysts can be compared with two varied measures? Is accuracy “3.99 mA cm-2” reliable one?
  6. What is the reason for different slope of N2 adsorption-desorption isothermal curve for PCZ-3 compared to all other specimens?
  7. From Figure 2D looks like Cu and Zn are within one particle. Did you consider the possibility of formation of Cu-Zn compounds and Pd particles separately?
  8. The authors stated that PCZ contains metallic Pd, Cu, Zn, whereas BM-3 only CuO and ZnO. What is the reason for that?
  9. Were all PXRD reflections indexed using metallic components and BTC? Did you not identified oxides in case of PCZs?
  10. In the manuscript the presence of elemental Pd, Cu and Zn are mentioned and illustrated with TEM. However, according Fig. S4 (A-C) Pd is mainly in oxidic form as well as Cu, whereas for Zn in overall no metallic Zn was found. It is clear discrepancy between experimental data and need careful re-consideration and clarification in the text.
  11. The authors mentioned shift of XP core levels of Pd 3d and Cu 2p towards higher BE and explained this by electron transfer between these components. If it so, why BEs are shifted in the same direction for both components? Are those shifts not due to the present oxides?
  12. The enhanced ECR activity is assigned to electron transfer between Cu and Zn. Do you have any proof of that? Is improved activity not due to higher SA?
  13. Which criteria were used for onset potential determination? Why potential -1.4 V vs. SCE was chosen for CA experiment?
  14. What is the accuracy of ECSA determination using capacitance method in your study?
  15. Was N2 adsorption-desorption performed for PCZ-2 after 10 h CA? The morphology changes and large crystallites become visible!
  16. The XPS after 10 h CA clearly shows the different ratios between metallic and oxidic contributions (compared to pristine materials). Please highlight and explain this in the main text.
  17. Is it possible to compare the ECR activity of PCZ-2 with other Cu-Zn electrocatalysts from literature? What is the advantage of MOF use? Were Pd-Cu-Zn also already investigated?
  18. Some comments and remarks concerning formatting of written manuscript:
  • Line 61: 15 m à 15 min;
  • Was pH measured? If yes, it will be very helpful to have this value along the Table S1 (or in the main text);
  • It is reasonable to add the insets to PXRD highlighting the presence of elemental Pd, Cu and Zn (it is important information and should be not hidden in SI).
  • It is necessary to add legends to Fig. S3 (upper pictures). It is not clear that doublet is presented separately. It is also not clear that ZnO(?) was fitted for XPS spectra of Zn 2p core levels.
  • In the main text CA is presented for 20 min, whereas there is very important data (10 h CA) with GC data in SI. Is it not reasonable to include these data in the main text? It is more relevant as short-term CA study.
  • Figures 4B, C are fully not informative à need improvement (normalization? Reference peaks?)
  • Please check the correctness of Refs.: 41, 44, 45.

Reviewer 2 Report

The manuscript entitled “Highly CO selective trimetallic metal-organic framework electrocatalyst for the electrochemical reduction of CO2” presents a trimetallic MOF option for the efficient CO2 electrocatalytic reduction reaction. The MOF choice made by the authors is one of the most exploited MOF networks (ZIF-8) and the preparation methods used to obtain PCZ-1 to 3 changes completely the initial structural features of this MOF, judging by the changes on the powder XRD. This is where a few doubts arise from the final materials. For instance:

1) In the powder XRD patterns the authors compare the most prominent diffraction peaks with Cu-BTC. However, no comparison is made with the simulated ZIF-8 diffractometer. The simulated diffractometers of both ZIF-8 and Cu-BTC (or HKUST) should be introduced in figure 3.

2) In reference 31 ZIF-8 is used as a sacrificial template and the authors do not mention this preparation characteristic in this manuscript. Please add this information to the manuscript and provide a few insight into the reasons for using this methodology.

3) The increase in Palladium probably disrupts the formation of Cu-BTC and can also be a reason for the differences observed in terms of electrocatalysis results amongst PCZ-2 and PCZ-3. Please discuss these aspects and clarify them in the manuscript.

4) Interestingly the use of three different metals in the network shown an extraordinary increment in the CO2 electrocatalytic reduction reaction. The authors explain this in terms of Palladium amount increase in the network. However, the PCZ-3 with a ratio of metal Pd/Cu/Zn of 3:1:1 seems to be less efficient than a ratio of 1:1:1 used in PCZ-2. This shows that somehow there is a synergistic effect with the presence of the 3 metals in this precise ratio. On the other hand, this can be attributed to the structural features of the materials obtained. The possible presence of a more ordered Cu-BTC network can be associated with better electrocatalytic results. Please clarify.

5) The authors should compare these results with other MOF materials in the literature used for the CO2 electrocatalytic reduction reaction.

Thank you

Round 2

Reviewer 2 Report

In general I am pleased with the authors responses to my concerns, and the general improvements made to the manuscript after the initial assessment. I am happy to accept the manuscript for publication in the current form.

Author Response

In general I am pleased with the authors responses to my concerns, and the general improvements made to the manuscript after the initial assessment. I am happy to accept the manuscript for publication in the current form.

--> Thank you for your valuable comments.